# Effect of Hindered Phenol Crystallization on Properties of Organic Hybrid Damping Materials

**DOI:** 10.3390/ma12071008

**Published:** 2019-03-27

**Authors:** Lin Zhang, Duoli Chen, Xiaoqiang Fan, Zhenbing Cai, Minhao Zhu

**Affiliations:** 1Key Laboratory of Advanced Technologies of Materials (Ministry of Education), School of Materials Science and Engineering, Southwest Jiaotong University, Chengdu 610031, China; zhanglin@swjtu.edu.cn (L.Z.); zhuminhao@139.com (M.Z.); 2Tribology Research Institute, State Key Laboratory of Traction Power, Southwest Jiaotong University, Chengdu 610031, China; chenduoli@my.swjtu.edu.cn (D.C.); czb-jiaoda@126.com (Z.C.)

**Keywords:** organic hybrid damping material, hindered phenol, hydrogen bonding, crystallization

## Abstract

Organic hybrid damping materials have achieved sustainable development in recent years for superior damping properties due to the hydrogen bonding of hindered phenol. However, the aggregation and crystallization of hindered phenol in the matrix can lead to a sharp decline in material properties. Thus, a series of hindered phenol hybrid carboxylated nitrile rubber (XNBR) composites with different types and contents of hindered phenol were prepared by melt blending to study the effects of different hindered phenol on the properties of organic hybrid damping materials. A dynamic mechanical analyzer (DMA) and scanning electron microscope (SEM) were used to study the dynamic mechanical properties and cross-section morphology of composites. X-ray diffraction (XRD) was used to study the crystallization of hindered phenol. The results show that the properties of organic hybrid damping materials were affected by the structure of hindered phenol, and that hindered phenol molecules with a linear structure had better performances. The greater the number of hydrogen bonds between hindered phenol and the XNBR matrix, the more difficult it was for the hindered phenol to crystallize.

## 1. Introduction

Organic hybrid damping materials are recently developed class of damping material. Hindered phenols are an interesting type of nonmetallic stabilizer, and are extensively used antioxidants for organic hybrid damping materials. This is especially the case in the fields of mechanical and aerospace engineering, as vibration and noise have become two of the key technical problems that must be solved. The multi-formant response of structures caused by broadband random excitation can make electronic devices and other instrumentation fail, and can cause the life of mechanical parts to shorten and then fail in serious cases, resulting in disastrous consequences [1,2,3,4,5,6]. Hindered phenol can be used alone to prevent polymer chain degradation, reduce the change in polymer color, and minimize the consumption of the hindered phenols during processing, thereby realizing the goal of long-term service life for the final products [7,8]. The damping property of the hybrid material is also related to the hydrogen bonds between the polymer and the small molecules [9,10]. When hydrogen bonds are formed between the polymer and the small molecules, a hydrogen bond network will be formed in the hybrid system, and under the influence of external forces, more energy can be consumed by hydrogen bond breaking and recombination [11,12]. The hybridization of small molecules and the polymer may change the state of the polymer and influence the rearrangement of molecular chain segments. Yin et al. studied the dynamic mechanical properties of AO-80 and chlorinated polyethylene (CPE) hybrids, and found that the improvement of dynamic properties was mainly attributed to the intermolecular hydrogen bonds between AO-80 and CPE [13]. Liu et al. found that the damping properties could be remarkably improved by introducing AO-2246 into carboxylated nitrile rubber (XNBR) [14]. A great deal of research has been conducted to improve the damping properties of damping materials through the introduction of small molecules [15,16,17].

Liu discussed the role of the number of hydrogen bonds in the hybrid system, and it was found that for the same small molecule, the greater the number of hydrogen bonds that formed between the small molecules and polymers, the stronger the damping effect of the small molecules on the polymers [18]. Cao et al. studied the effect of hindered phenol crystal morphology on the mechanical properties of organic hybrid materials [19]. In recent years, hydrogen bonds have been the most attractive phenomenon to those fields of chemistry, physics, and biology [20]. Many experimental methods have been used to investigate hydrogen bonds in an indirect and qualitative way. Song et al. quantitatively studied the microscopic structure and hydrogen bonds of AO-60/NBR composites using molecular dynamics (MD) simulation [21]. MD simulation is always used to explore the underlying damping mechanism and the relationship between the hydrogen bonds and the damping properties at the molecular level [22]. With the growth of extremely powerful computer hardware and software in decades past, MD simulation can reveal the structure–performance relationships of materials as a powerful theoretical tool [23,24]. In many studies, the research regarding the relationship between the structure of the hindered phenol and dynamic mechanical and aging properties has not been sufficient. The corresponding research is urgently required [25].

In this study, compounds containing small molecules with different structures were prepared, and the dynamic mechanical properties of hindered phenol were investigated. The crystallization properties of small molecules and hydrogen bonds were characterized. The difference of hydrogen bonds between different structures of hindered phenolic materials was simulated by molecular dynamics, and the effect of hydrogen bonds on the properties of organic hybrid damping materials was studied.

## 2. Experimental Procedures

### 2.1. Materials

XNBR (1027) with an acrylonitrile mass fraction of 27% was provided by NANTEX Industry Co., Ltd. (Kaohsiung, Taiwan). Meanwhile, AO-2246 (2,2’-methylenebis(6-tert-butyl-4-methyl-phenol), AO-80 (3,9-bis[1,1-dimethyl-2-[(3-tert-butyl-4-hydroxy-5-methylphenyl)propionyloxy]ethyl]-2,4,8,10-tetraoxaspiro), AO-60 (2,2-bis[3-(3,5-di-tert-butyl-4-hydroxyphenyl)propionyloxymethyl] triethylene), and XH-245 (triethylene glycol bis(3-tert-butyl-4-hydroxy-5-methylphenyl)propionate) powders were purchased from BASF SE (Ludwigshafen, Germany). The role that these hindered phenols play in the matrix has been explored to a great extent in previous studies [26]. Researchers have found that XH-245 is a hindered phenol with a long chain, which can also form strong intermolecular interactions with NBR. Other chemicals and ingredients were purchased in Kelong (Chengdu, China). All materials were used without further purification.

### 2.2. Preparation of the Composites

Hindered phenol/XNBR composites were prepared in the following procedures: (a) XNBR was kneaded by a two-roll mill at room temperature for 2 min; (b) hindered phenols were added and then kneaded at room temperature for 5 min, and their mass ratios were as shown in Table 1; (c) the composites were hot-pressed at 160 °C under a pressure of 15 MPa for different periods of time, and then cooled down in an ice water bath before quenching to prepare the hindered phenol/XNBR samples.

### 2.3. Characterization

The dynamic mechanical properties of the samples were measured by a Q800dynamic mechanical analyzer (DMA) (TA Instruments, New Castle, DE, USA, stretching mode, test frequency was 1 Hz, Poisson ratio was 0.44), XRD (AXIS-ULTRA, scan range was 60°, scanning rate was 5° per minute) and SEM (FEI, Hillsboro, OR, USA, QUANTA200-SEM, accelerating voltage was 10 V). Chemical groups of samples were measured using FT-IR with a KBr pellet (FT-IR, Nicolet670, Nicolet, Madison, WI, USA, ATR mode, scanning rate was 1 cm/s).

## 3. Results and Discussion

### 3.1. Aging Properties of Hindered Phenolic Organic Hybrid Materials

The aging test was conducted at a temperature of 60 °C, with 4 h aging time. In order to avoid the deterioration of the original sample, the adopted samples were tested within 2 h of the preparation, and the remaining samples were stored at low temperature. The varying performance of different hindered phenol organic hybrid damping materials before and after aging tests can be seen in Figure 1, Figure 2, Figure 3, Figure 4 and Figure 5. The findings suggested that adding a hindered phenol into XNBR can significantly increase the loss factor of the material, while the glass transition temperature of the compound moved to a higher temperature. The results demonstrated that with the increase of AO-2246 (Figure 1), the loss factor increased from 1.2 to 1.7, and meanwhile the glass transition temperature and the storage modulus also rose from 0 to about 18 °C and 700 to 2000 MPa, respectively. After the 60 °C/4 h aging process, the glass transition temperature and the loss factor of the compounds decreased to below 0 °C and about 0.8, respectively, with the storage modulus reduced to less than 1000 MPa. However, samples with AO-80 had different performances, such that with the increase of AO-80 (Figure 2), the glass transition temperature and the loss factor were increased from 3 °C to 18 °C and 1.22 to 1.74, respectively, and the storage modulus increased from 500 to 1350 MPa. Thus, it can be concluded that aging had almost no effect on storage modulus or loss factor. AO-60 and XH-245 were also selected. Figure 3 shows the performance of compounds with AO-60. It can be seen that with the increase of AO-60, the glass transition temperature and the loss factor were increased from 7 °C to 21 °C and 1.42 to 2.43, respectively, while the storage modulus was increased from 600 to 1180 MPa. After the aging test, the loss factor and the storage modulus were reduced to below 1.5 and 800 MPa, respectively. XH-245 had a similar varying trend with AO-80 (Figure 4). This means that with the increase of XH-245, the glass transition temperature and the loss factor were increased from 9 °C to 10 °C and 1.19 to 1.67 respectively, with the storage modulus increasing from 423 to 765 MPa. The aging test indicated that aging had almost no impact on the storage modulus or the loss factor. It can be seen from Figure 5 that the loss factor and the storage modulus of the samples with AO-2246 and AO-60 decreased substantially, whereas those of samples with AO-80 and XH-245 were almost unaffected by aging (Figure 5).

The high damping of organic hybrid materials was derived from hydrogen bonds. Hydrogen bonds not only form between the organic small molecules and the rubber molecular chains, but also between the small molecules in compounds. Damping properties can be enhanced by the process of breaking and recombining two types of hydrogen bonds, whereas the hydrogen bonds between the small organic molecules would promote the aggregation and crystallization of small molecules.

### 3.2. Crystallization Properties of the Hindered Phenol in Compounds

It could be seen in the damping performance characterization that compounds of different hindered phenols had different properties during the aging test. Thus, XRD and SEM were used to study the crystallization of different hindered phenols in compounds. XRD results indicate that under the condition of unannealed treatment, there was only one peak in the sample. This basically shows the characteristic of the large amorphous peak, so most of the samples had amorphous structure. However, for annealed samples, there were strong diffraction peaks, which indicates that a large number of ordered structures were generated inside the samples. Moreover, this ordered structure was characterized by the formation of the crystal (Figure 6).

The cross sections of samples with different hindered phenol are shown in Figure 7, Figure 8, Figure 9 and Figure 10. Many crystals were observed on the sections of the samples with AO-2246, AO-60, and XH-245 after the aging test. The amount of crystallization increased with the increase of hindered phenol content. It can be seen from the fracture appearance of the sample with AO-2246 that a large number of fine crystals were observed in the section, and the cross section shows that the XNBR matrix had good compatibility with AO-2246 before the annealing test, whereas crystallization of the hybrid phenol resulted from thermal oxidation. Similar performance can be noticed in the sample with AO-60, with fine crystals spreading throughout the cross section. The compatibility of the XNBR matrix with AO-80 was better, and even after the aging test, the cross section was still smooth without crystals. XRD showed a small number of ordered structures inside the sample. When the AO-80 content was high, similar large crystals were observed on the surface. Large crystals were observed in the cross sections of samples with XH-245. Combining this with the loss factor and the storage modulus analysis, it can be noted that crystallization did not undermine the performance of damping materials when the content of XH-245 was low, which was mainly because of the formation of XH-245 crystals with a wide sheet structure.

### 3.3. Mechanism of Organic Hybrid Crystallization

The MD method has been one of the most effective for studying small molecular hybrid damping materials in recent years. The MD simulation was carried out by using the Discover and Amorphous Cell modules from Accelrys company. The initial velocities of the atoms were assigned by using the Maxwell–Boltzmann profiles at 298 K, the Verlet velocity time integration method was used with the time step of 1 femtosecond, the energy of each cell was minimized to 1.0 × 10^−5^ kcal/(mol·A), and then the cell was annealed at 0.1 MPa from 200 to 400 K. Then, 200 ps of NVT at 298 K and 200 ps of NPT simulation was performed at 0.1 MPa to further relax. The results of the previous experiments indicate that the hydrogen bond between molecules in Figure 11 may be the main factor in the crystallization of the hindered phenol small molecules, and it is also a key factor for the declining properties of small molecular hybrid materials. In order to study the relationship between hydrogen bonds and hindered phenol crystallization, a theoretical model was set up according to the method in the literature [19,20]. The model was annealed and processed through the NPT and NVT ensemble, and the system was stable at 258 K and fluctuated no more than 10 K, indicating that the system had reached an equilibrium. The number of hydrogen bonds in the crystal cell was counted, and the result is listed in Table 1. With the increase of hindered phenol content, the number of hydrogen bonds increased. From the simulation results, it can be noted that there were more of hydrogen bond (B) than hydrogen bond (A) in XNBR with AO-2246, which indicates that the reason for the high damping of XNBR/AO-2246 was the hydrogen bonding between AO-2246 molecules. Crystallized of AO-2246 is also the reason for the degradation of the material properties. Similar results for AO-60 are shown in Table 2. The experiments conducted for samples with AO-80 had different results. More of hydrogen bond (A) were found than hydrogen bond (B) in samples with AO-80, indicating that the reason for the high damping of XNBR/AO-80 was the hydrogen bonding between AO-80 and XNBR. When the content of XH-245 was low, the hydrogen bonds of the mixture were dominated by hydrogen bond (A), and the amount of hydrogen bonds between XH-245 molecules increased with the increase of the content of XH-245. As the XH-245 crystallized, its damping performance worsened.

### 3.4. Discussion

According to the above analysis and results, we put forth that the reason for the high damping capacity of organic hybrid damping materials is hydrogen bonding, and that there are two types of hydrogen bonds formed by hindered phenols in the matrix. From the FT-IR results, it can be seen that there were hydrogen bonds in samples before the aging tests (peak between 3000 and 3600 cm^−1^) in Figure 12, and the hydrogen bonds in XH-245, AO-60, and AO-2246 disappeared after annealing. The difference in the crystallization properties of different hindered phenols was also one of the reasons leading to the difference in material properties. Moreover, the crystallization of the hindered phenols can lead to the disappearance of hydrogen bonds in the material system, and thus the crystal formation would also degrade the material properties. However, it was found that not all hindered phenols had similar properties. We found that the properties and durability of organic hybrid materials were determined by the type of hydrogen bond in the matrix, the hindered phenol crystallization properties, and crystal morphology altogether. AO-80 and XH-245 are molecules with linear structure and have flexible chain characteristics, making crystallization more difficult for them. AO-80 and XH-245 became amorphous in the process of sample preparation, forming a hydrogen bonding network in the mixture. The difference in crystallization properties of several hindered phenols was found in XRD results. Meanwhile, it can be seen that the crystallization exothermic peaks of AO-2246 and AO-60 appeared at 82.28 °C and 46.34 °C, respectively. However, no crystallization peak occurred for AO-80 and XH-245. This indicates that linear hindered phenolic molecules have certain advantages in prolonging the service life of organic hybrid damping materials.

## 4. Conclusions

According to the experimental results, the following conclusions can be obtained:(1)The high damping capacity of organic hybrid damping materials is attributed to hydrogen bonding, and the crystallization of hindered phenols can lead to the disappearance of hydrogen bonds in the material system. Thus, the crystals can decrease the material properties.(2)The molecular structure of the hindered phenol had an influence on the properties of organic hybrid damping materials, and hindered phenol molecules with linear and branching structure had better performances.(3)The molecular structures of AO-2246 and AO-60 are more symmetrical and regular than that of linear hindered phenols, which makes crystallization easier. There are more hydrogen bonds between the hindered phenol itself in the matrix, thus promoting agglomeration and crystallization.(4)AO-80 and XH-245 had good crystallization resistance properties. There was a greater number of hydrogen bonds between the hindered phenol and XNBR matrix. When XH-245 was crystallized at a higher content, the crystal had a lamellar structure similar to that of sheet graphite, which was conducive to maintaining the material properties.

## Figures and Tables

**Figure 1 materials-12-01008-f001:**
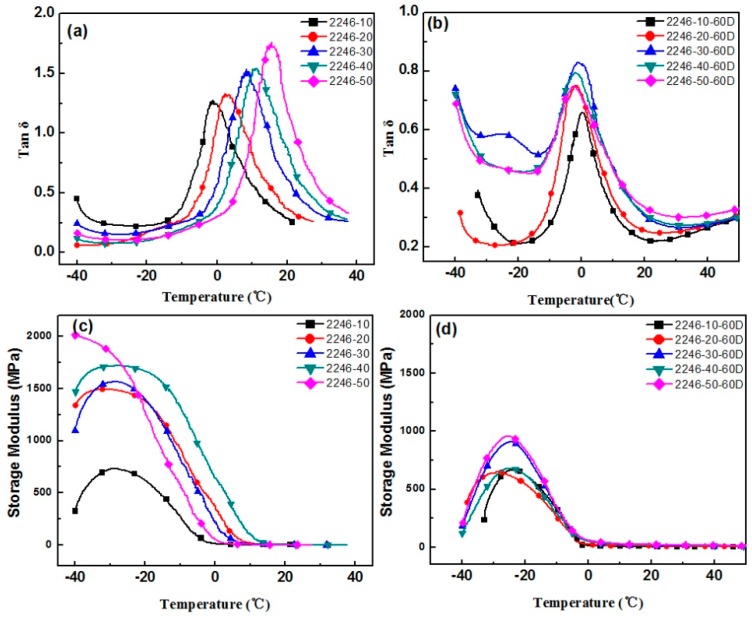
The effect of thermal aging on the storage modulus and tan δ of the carboxylated nitrile rubber (XNBR)/AO-2246 blends: (**a**) tan δ of XNBR/AO-2246; (**b**) tan δ of XNBR/AO-2246 after aging; (**c**) the storage modulus of XNBR/AO-2246; (**d**) the storage modulus of XNBR/AO-2246 after aging.

**Figure 2 materials-12-01008-f002:**
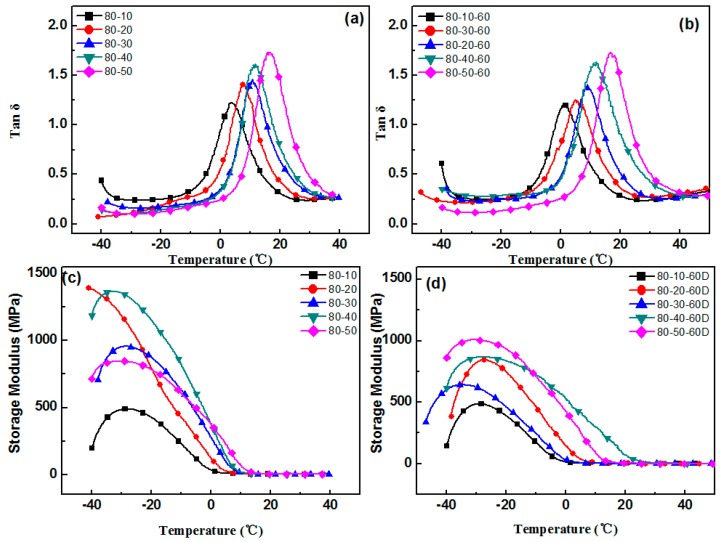
The effect of thermal aging on the storage modulus and tan δ of the XNBR/AO-80 blends: (**a**) tan δ of XNBR/AO-80; (**b**) tan δ of XNBR/AO-80 after aging; (**c**) the storage modulus of XNBR/AO-80; (**d**) the storage modulus of XNBR/AO-80 after aging.

**Figure 3 materials-12-01008-f003:**
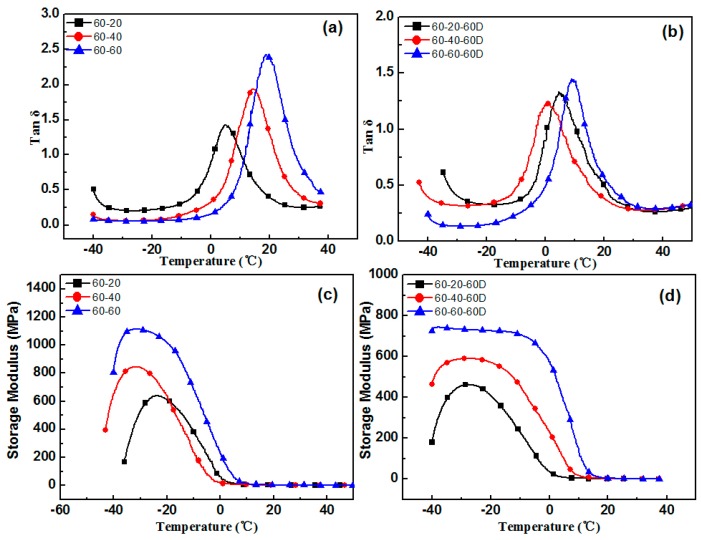
The effect of thermal aging on the storage modulus and tan δ of the XNBR/AO-60 blends: (**a**) tan δ of XNBR/AO-60; (**b**) tan δ of XNBR/AO-60 after aging; (**c**) the storage modulus of XNBR/AO-60; (**d**) the storage modulus of XNBR/AO-60 after aging.

**Figure 4 materials-12-01008-f004:**
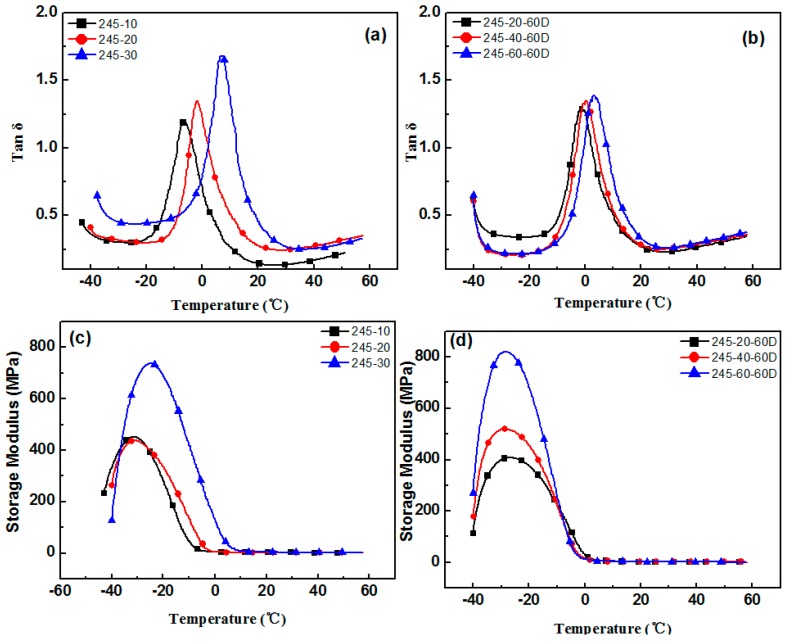
The effect of thermal aging on the storage modulus and tan δ of the XNBR/XH-245 blends: (**a**) tan δ of XNBR/XH-245; (**b**) tan δ of XNBR/XH-245 after aging; (**c**) the storage modulus of XNBR/XH-245; (**d**) the storage modulus of XNBR/XH-245 after aging.

**Figure 5 materials-12-01008-f005:**
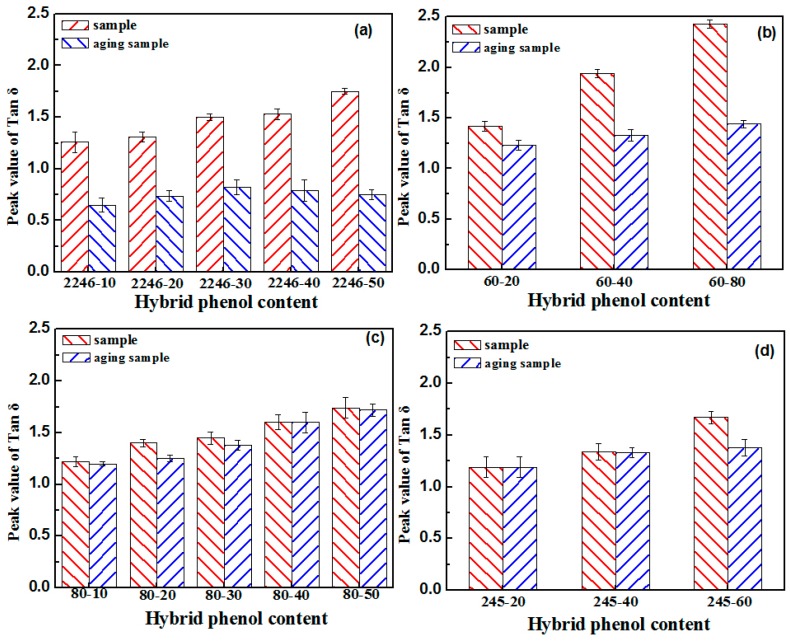
Comparison of the peak of tan δ of XNBR/hybrid phenol blends after thermal aging (**a**) Peak value of tan δ of XNBR/AO-2246; (**b**) Peak value of tan δ of XNBR/AO-60; (**c**) Peak value of tan δ of XNBR/AO-80; (**d**) Peak value of tan δ of XNBR/XH-245.

**Figure 6 materials-12-01008-f006:**
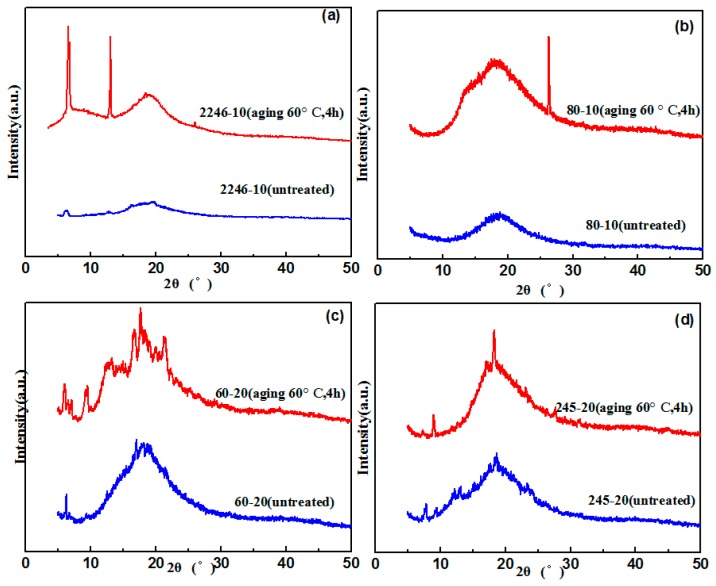
XRD patterns of XNBR/hybrid phenol blends (**a**) XRD patterns of XNBR/AO-2246; (**b**) XRD patterns of XNBR/AO-80; (**c**) XRD patterns of XNBR/AO-60; (**d**) XRD patterns of XNBR/XH-245.

**Figure 7 materials-12-01008-f007:**
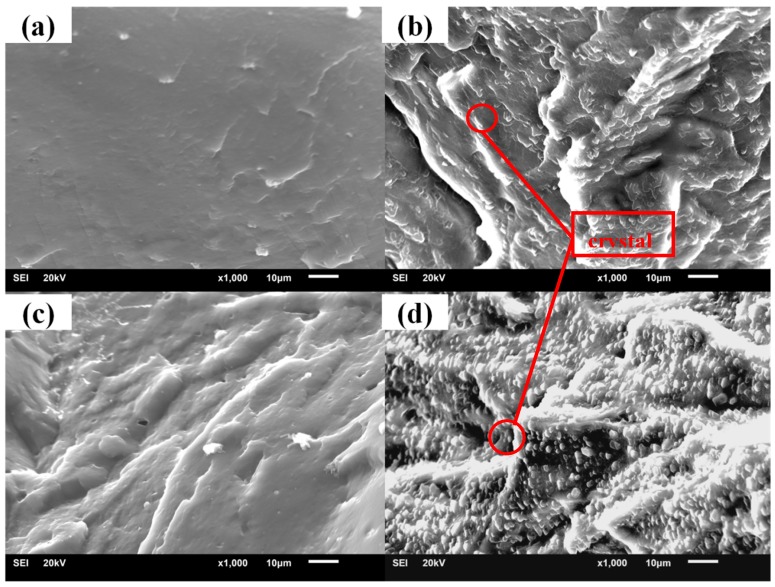
SEM patterns of XNBR/AO-2246 before and after thermal aging: (**a**) XNBR/AO-2246(10); (**b**) XNBR/AO-2246(10) after aging; (**c**) XNBR/AO-2246(50); (**d**) XNBR/AO-2246(50) after aging.

**Figure 8 materials-12-01008-f008:**
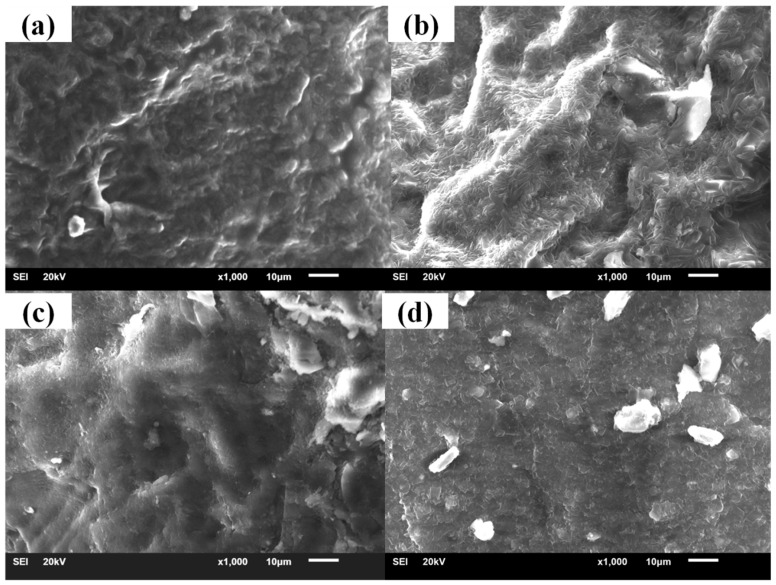
SEM patterns of XNBR/AO-60 before and after thermal aging: (**a**) XNBR/AO-60(20); (**b**) XNBR/AO-60(20) after aging; (**c**) XNBR/AO-60(60); (**d**) XNBR/AO-60(60) after aging.

**Figure 9 materials-12-01008-f009:**
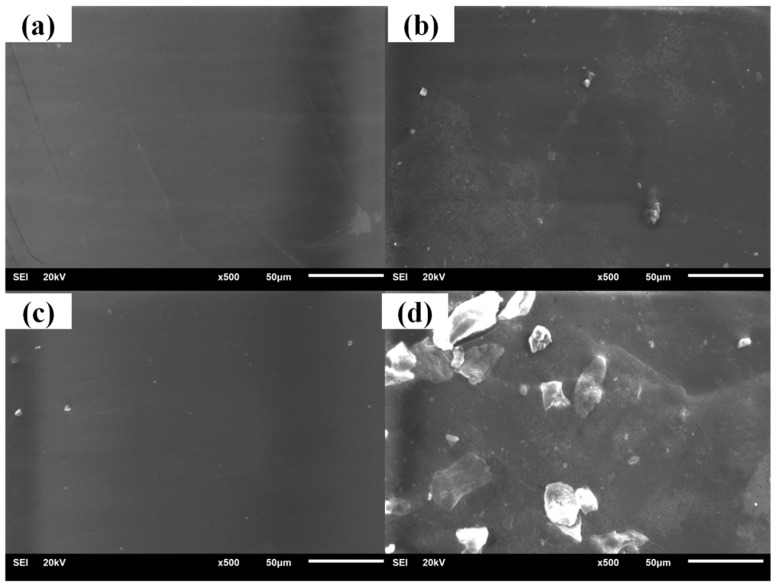
SEM patterns of XNBR/AO-80 before and after thermal aging: (**a**) XNBR/AO-80(10); (**b**) XNBR/AO-80(10) after aging; (**c**) XNBR/AO-80(50); (**d**) XNBR/AO-80(50) after aging.

**Figure 10 materials-12-01008-f010:**
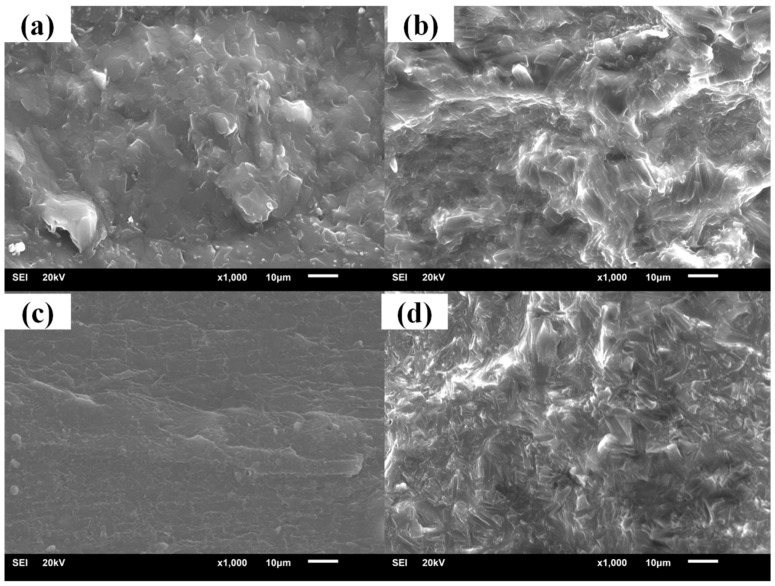
SEM patterns of XNBR/XH-245 before and after thermal aging: (**a**) XNBR/XH-245(20); (**b**) XNBR/XH-245(20) after aging; (**c**) XNBR/XH-245(60); (**d**) XNBR/XH-245(60) after aging.

**Figure 11 materials-12-01008-f011:**
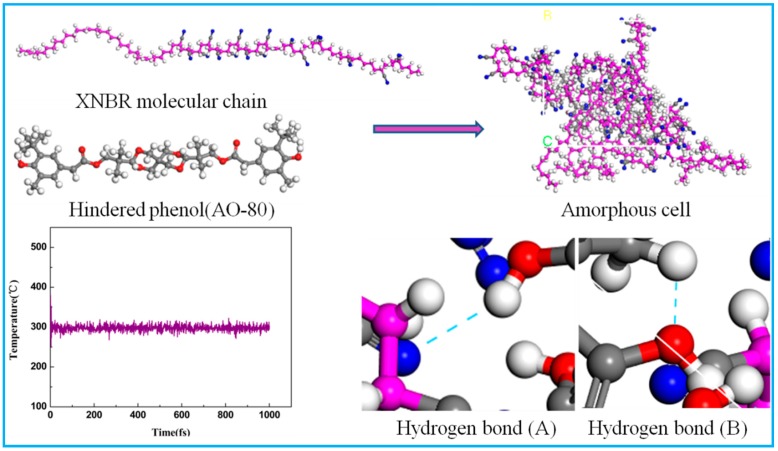
Models for molecular dynamics simulation of XNBR/hybrid phenol composites.

**Figure 12 materials-12-01008-f012:**
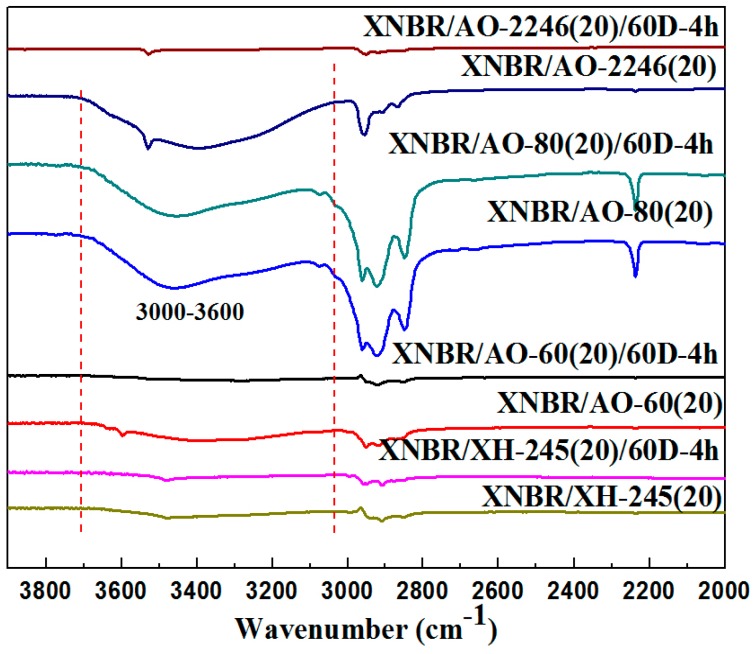
FT-IR spectrum of XNBR/hybrid phenol composites.

**Table 1 materials-12-01008-t001:** Material formulations of rubber compounds.

Materials	Content (Parts Per Hundreds of Rubber)			
	XNBR/AO-80	XNBR/AO-2246	XNBR/AO-60	XNBR/XH-245
XNBR	100	100	100	100
AO-80	10/20/30/40/50	0	0	0
AO-2246	0	10/20/30/40/50	0	0
AO-60	0	0	20/40/60	0
XH-245	0	0	0	20/40/60

**Table 2 materials-12-01008-t002:** Number of hydrogen bonds (A) and hydrogen bonds (B) in different XNBR/hybrid phenol composites.

Samples	Content	Hydrogen Bond (A)	Hydrogen Bond (B)
AO-2246	XNBR/AO-2246(10)	1	2
	XNBR/AO-2246(20)	2	4
	XNBR/AO-2246(30)	2	4
	XNBR/AO-2246(40)	3	5
	XNBR/AO-2246(50)	3	6
AO-80	XNBR/AO-80(10)	1	0
	XNBR/AO-80(20)	2	0
	XNBR/AO-80(30)	2	0
	XNBR/AO-80(40)	3	1
	XNBR/AO-80(50)	4	2
AO-60	XNBR/AO-60(20)	2	1
	XNBR/AO-60(40)	4	2
	XNBR/AO-60(60)	5	3
XH-245	XNBR/XH-245(20)	1	0
	XNBR/XH-245(40)	2	1
	XNBR/XH-245(60)	4	3

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
