# Peer review of "Effect of Hindered Phenol Crystallization on Properties of Organic Hybrid Damping Materials"

_materials, 2019, doi:10.3390/ma12071008_

Reviewer 1 Report

The paper is devoted to the current issues of the development of organic hybrid damping materials based on polymer matrix with superior damping property due to the hydrogen bonding of hindered phenol. The dynamic mechanical properties and cross section morphology of a series of hybrid carboxylated nitrile rubber/hindered phenol composites with different type and different contents of hindered phenol as well as the crystallization process of hindered phenol were investigated. The results presented in this paper are interesting and presented at an acceptable level. However, I believe that my comments below could help to increase the reader's communication potential:

Abbreviations for antioxidants and inhibitors AO-2246, AO-80, AO-60 and XH‐245 by should be explained (line 67).

Line 68 – add some references to previous studies referred by the authors.

More detailed description of test conditions (DMA, XRD, SEM and FT-IR) is needed.

Add missing x-axis description to the Figure no. 5.

The manuscript may become acceptable after minor revisions of content. This manuscript can have a significant impact on further material research and development.

Author Response

1. Abbreviations for antioxidants and inhibitors AO-2246, AO-80, AO-60 and XH‐245 by should be explained (line 67).

Response 1: Abbreviations of the antioxidants has been explained.

2. Line 68 – add some references to previous studies referred by the authors.

Response 2: The previous studies referred has been added.

3. More detailed description of test conditions (DMA, XRD, SEM and FT-IR) is needed.

Response 3: More detailed of the test conditions has been added.

4. Add missing x-axis description to the Figure no. 5.

Response 4: X-axis descrription has been added, thank you for your work.

Reviewer 2 Report

In this paper, the influence of the hindered phenol crystallization on the mechanical and chemical properties of organic hybrid damping materials is experimentally and numerically investigated. A dynamic mechanical analyser and a scanning electron microscope are used to study the dynamic mechanical properties and the cross section morphology of the composites, respectively. The Xray diffraction method is used to observe the crystallization of the hindered phenol. Molecular dynamics simulations are carried out to study the effect of different hydrogen bonds on the properties of the hindered phenolic materials.

The paper is well written and arranged, where the results of the experimental and numerical studies are rightly reported and illustrated. In particular, they are very interesting and clear the SEM patterns of Figures 7-10.

In order to consider this paper for publication, there are only two relevant aspects that should be carefully taken into account by the Author.

1) The quality of Figures 1-6 should be improved: this figures appear to be just copied and pasted PDF images, and not vector-type images.

2) In the Introduction, they should be emphasized the applications of the damping materials in the industrial fields, in particular the mechanical and aerospace fields. For example, the Authors should underline the role of the damping materials in order to obtain reduced mechanical vibrations by means of their use in multi-layer composites. Therefore, the following relevant papers should be added and discussed:

a) Catania, G.; Strozzi, M. Damping oriented design of thin-walled mechanical components by means of multi-layer coating technology. Coatings 2018, 8, 73.

b) Yu, L.; Ma, Y.; Zhou, C.; Xu, H. Damping efficiency of the coating structure. International Journal of Solids and Structures 2005, 42, 3045-3058.

c) Rongong, J.A.; Goruppa, A.A.; Buravalla, V.R.; Tomlinson, G.R.; Jones, F.R. Plasma deposition of constrained layer damping coatings. Proceedings of the Institution of Mechanical Engineers, Part C: Journal of Mechanical Engineering Science 2004, 218, 669-680.

Other industrial applications of the damping materials would be very useful for the scientific completeness of the study presented in the paper.

Therefore, in the opinion of the Reviewer, by considering the previous notes, the paper should be accepted for publication after minor revisions.

Author Response

1) The quality of Figures 1-6 should be improved: this figures appear to be just copied and pasted PDF images, and not vector-type images.

Response 1: Figures 1-6 has been corrected.

2) In the Introduction, they should be emphasized the applications of the damping materials in the industrial fields, in particular the mechanical and aerospace fields. For example, the Authors should underline the role of the damping materials in order to obtain reduced mechanical vibrations by means of their use in multi-layer composites. Therefore, the following relevant papers should be added and discussed:

a) Catania, G.; Strozzi, M. Damping oriented design of thin-walled mechanical components by means of multi-layer coating technology. Coatings 2018, 8, 73.

b) Yu, L.; Ma, Y.; Zhou, C.; Xu, H. Damping efficiency of the coating structure. International Journal of Solids and Structures 2005, 42, 3045-3058.

c) Rongong, J.A.; Goruppa, A.A.; Buravalla, V.R.; Tomlinson, G.R.; Jones, F.R. Plasma deposition of constrained layer damping coatings. Proceedings of the Institution of Mechanical Engineers, Part C: Journal of Mechanical Engineering Science 2004, 218, 669-680.

Response 2: The relevant papers has been added and discussed in introduction.

Reviewer 3 Report

 Journal title: Materials  - paper 468708

Effect of hindered phenol crystallization on properties of organic hybrid damping materials

Lin Zhang, Duoli Chen, Xiaoqiang Fan, Zhenbing Cai, Minhao Zhu

Comments to the pictures:

Figure1 d: Storage modulus of XNBR/AO‐2246 after aging. You can increase the scale of Storage Modulus to 1.000 MPa.

The rest figures and SEM photo are OK.

Comments to the paper text:

Row 37:  is “studisd” should be studied

Row 177: is “1femtosecond”  better 1 fem to second

References: You should add some more actual papers from 2017-2019 years. Now it’s only 2 from 2017 year.

General comments:

Very interesting and important article, well chosen research materials, possible usefulness of results in industrial practice, very clearly presented research. Good research description and results analysis. Interesting models for MD simulation of XNBR/hybrid phenol composites (figure 11). After minor revision article worth publishing in the Materials Journal.

Author Response

Comments to the pictures:

Figure1 d: Storage modulus of XNBR/AO‐2246 after aging. You can increase the scale of Storage Modulus to 1.000 MPa.

The rest figures and SEM photo are OK.

 Response 1:The Figure has been corrected.

Comments to the paper text:

Row 37:  is “studisd” should be studied

Row 177: is “1femtosecond”  better 1 fem to second

References: You should add some more actual papers from 2017-2019 years. Now it’s only 2 from 2017 year.

 Response 2: Errors in the article has been corrected, documents in recent years have also been added.
